# Antibiotic Prescribing in Dutch Daytime and Out-of-Hours General Practice during the COVID-19 Pandemic: A Retrospective Database Study

**DOI:** 10.3390/antibiotics11030309

**Published:** 2022-02-25

**Authors:** Karin Hek, Lotte Ramerman, Yvette M. Weesie, Anke C. Lambooij, Maarten Lambert, Marianne J. Heins, Janneke M. T. Hendriksen, Robert A. Verheij, Jochen W. L. Cals, Liset van Dijk

**Affiliations:** 1Nivel, Netherlands Institute for Health Services Research, P.O. Box 1568, 3500 BN Utrecht, The Netherlands; l.ramerman@nivel.nl (L.R.); y.weesie@nivel.nl (Y.M.W.); m.heins@nivel.nl (M.J.H.); j.hendriksen@nivel.nl (J.M.T.H.); r.verheij@nivel.nl (R.A.V.); l.vandijk@nivel.nl (L.v.D.); 2IVM, The Dutch Institute for Rational Use of Medicine, P.O. Box 3089, 3502 GB Utrecht, The Netherlands; a.lambooij@ivm.nl; 3Unit of PharmacoTherapy, Groningen Research Institute of Pharmacy, Epidemiology & Economics, University of Groningen, P.O. Box 72, 9700 AB Groningen, The Netherlands; m.lambert@rug.nl; 4Tranzo, Tilburg School of Social and Behavioral Sciences, Tilburg University, P.O. Box 90153, 5000 LE Tilburg, The Netherlands; 5Department of Family Medicine, Maastricht University, P.O. Box 616, 6200 MD Maastricht, The Netherlands; j.cals@maastrichtuniversity.nl

**Keywords:** COVID-19, restrictions, antibiotics, antimicrobial resistance, general practice, out-of-hours

## Abstract

COVID-19 restrictions have resulted in major changes in healthcare, including the prescribing of antibiotics. We aimed to monitor antibiotic prescribing trends during the COVID-19 pandemic in Dutch general practice, both during daytime and out-of-hours (OOH). Routine care data were used from 379 daytime general practices (DGP) and 28 OOH-services over the period 2019–2021. Per week, we analyzed prescription rates per 100,000 inhabitants, overall, for respiratory and urinary tract infections (RTIs and UTIs) specifically and within age categories. We assessed changes in antibiotic prescribing during different phases of the pandemic using interrupted time series analyses. Both at DGPs and OOH-services significantly fewer antibiotics were prescribed during the COVID-19 pandemic after government measures became effective. Furthermore, the number of contacts decreased in both settings. When restrictions were revoked in 2021 prescription rates increased both at DGP and OOH-services, returning to pre-pandemic levels at OOH-services, but not in DGP. Changes in antibiotic prescribing rates were prominent for RTIs and among children up to 11 years old, but not for UTIs. To conclude, while antibiotic prescribing decreased during the first year of the COVID-19 pandemic both in daytime and out-of-hours, the pandemic does not seem to have a lasting effect on antibiotic prescribing.

## 1. Introduction

Worldwide, the COVID-19 pandemic and the government measures implemented to reduce the spread of the SARS-CoV-2 virus have led to major changes in people’s lives and put great pressure on healthcare systems [1]. Healthcare providers, including general practitioners (GPs), had to reorganize their working routines. Face-to-face consultations were partially replaced by remote consultations [2], and in many countries, the number of contacts greatly reduced during the first months of the pandemic, both during daytime general practice (DGP) and out-of-hours (OOH) [1,3]. Furthermore, infection transmission, in general, decreased, due to social distancing, improved hand hygiene, and closing of day-care facilities and schools, contributing to a lower incidence of infectious diseases, including respiratory infections [4,5,6].

All these changes may have impacted the prescription of antibiotics [4]. For example, it has been suggested that the change from face-to-face consultations to remote consultations may have led to an increase in the prescribing of antibiotics, to overcome diagnostic uncertainty caused by the inability to perform physical examinations during remote consultations [7,8]. However, similar or less antibiotic prescribing during remote consultations has been observed as well [9,10,11]. Nevertheless, remote consultations where no antibiotic was prescribed may have been followed by face-to-face follow-up consultations. In addition, there could be a difference in antibiotic prescribing during remote consultations between diagnoses, i.e., studies show that antibiotics may be prescribed more frequently during remote consultations for urinary tract infections (UTIs) [11]. Moreover, since the start of the COVID-19 pandemic, a substantial decrease in antibiotic prescriptions was observed in most European countries on top of the already declining prescription rates over the last years [12]. This reduction was observed both in DGPs and OOH-services [10,13]. Some studies suggest that the reduction of antibiotic prescribing is linked to the reduced transmission of respiratory tract infections (RTIs) as this coincided with fewer prescriptions or dispensations of antibiotics for RTIs [10,14,15]. On the other hand, the antibiotic prescription rate for UTIs seems to stay comparable to pre-pandemic levels [15,16,17]. Furthermore, patients were discouraged to visit a GP when not strictly necessary, reducing the absolute number of contacts which may lead to a prescription of antibiotics [18,19,20,21].

Since the start of the COVID-19 pandemic, the impact of COVID-19 on society and the healthcare system has changed. Box 1 provides an overview of important changes in government measures and infection rates during the COVID-19 pandemic in the Netherlands. Restrictions to limit the spread of COVID-19 were eased and introduced again, general practice contact rates went back to normal, common infections started to spread again, as was observed for RSV [22,23,24], and COVID-19 vaccines were introduced. Several studies have shown the impact on antibiotic prescribing until the first months of the COVID-19 pandemic [10,13,15,17,25,26]. It is unknown whether this decrease sustained during 2021. Furthermore, the OOH setting has been hardly studied up till now [10,13], although antibiotics are one of the most commonly prescribed medications in this setting [27]. Therefore, in the current study, we aimed to describe (1) changes in antibiotic prescribing in the different phases of the COVID-19 pandemic in 2020 and 2021 compared to 2019 in DGPs and at OOH-services and (2) antibiotic prescribing for the two diagnoses for which antibiotics are most frequently prescribed, RTIs and UTIs, separately in relation to the different phases of the pandemic in the Netherlands.

Box 1COVID-19 and restrictive measures timeline in the Netherlands.For this study we identified the following phases of the COVID-19 pandemic in the Netherlands:**Phase 0** (week 1 2019–week 8 2020): pre-pandemic phase.**Phase 1** (week 9 2020–week 24 2020): first lockdown: following the first infections. The first COVID-19 patient in the Netherlands was reported on 27 February 2020. Measures included social distancing; closing of schools, child daycare, restaurants, and sports facilities; and working from home. Testing through municipality services was only available on a limited basis.**Phase 2** (week 25 2020–week 41 2020): Intermediate phase during the summer with fewer infections and fewer restrictions, i.e., schools, child daycare, restaurants, and sports facilities were open. Test capacity was increasing.**Phase 3** (week 42 2020–week 16 2021): Second lockdown following the second wave of infections with more strict lockdown measures, including social distancing, the closing of schools, child daycare, restaurants, sports facilities, non-essential stores, working from home, wearing face masks, and an evening clock. Testing was widely available. Vaccination was started in January 2021.**Phase 4** (week 17 2021–week 42 2021): Second intermediate phase without restrictions, except for basic advice regarding hand hygiene, social distancing, and testing. Testing was available for all, self-testing was introduced and the vaccination coverage was rapidly increasing.

## 2. Results

### 2.1. Trend in Antibiotic Prescribing in DGP and OOH-Services

Figure 1 shows the changes in antibiotic prescribing for all indications in DGP and OOH-services per week and Table 1 the average antibiotic prescribing rate during each of the phases in the pandemic.

Antibiotic prescribing in DGP during the first lockdown was significantly lower than in the pre-pandemic phase but showed high variation between the weeks (relatively large standard deviation, Table 1 and Figure 1). In addition, the antibiotic prescribing rate in DGP showed a decreasing trend during the two lockdowns (phase 1 and 3) and an increasing trend in the intermediate phases (phase 2 and 4, results of the interrupted time series analyses (ITSA) are shown in Appendix A). Nevertheless, the level of antibiotic prescribing in the second intermediate phase was still significantly lower than in the same period in 2019 (Appendix A).

Furthermore, in OOH-services antibiotic prescribing decreased significantly during the COVID-19 pandemic (Figure 1), but not until the second lockdown (Table 1). However, an increasing trend was observed during the second lockdown, compared to the preceding intermediate phase (phase 2, ITSA results in Appendix A), resulting in an increased prescription rate during the second intermediate phase (Table 1). While in DGP prescription rates during the second intermediate phase remained lower than in 2019, this was not the case for OOH-services: on average prescription rates did not differ between the second intermediate phase and the same period in 2019, before the COVID-19 pandemic (36.6/100,000 inhabitants, Appendix A). 

### 2.2. Trends for Respiratory Tract infections

Figure 2 and Figure 3 shows the number of contacts with and without an antibiotic prescription for RTIs per 100,000 inhabitants in DGPs and OOH-services, respectively, and the percentage of contacts for RTIs with an antibiotic prescription. 

In DGPs, the total number of contacts for RTIs showed a temporary increase during the first lockdown (phase 1; Figure 2), followed by a decrease and a continued lower level of contacts for RTIs during phases 2 to 4, than the pre-pandemic phase. The number of prescriptions for RTIs per 100,000 inhabitants decreased during the first lockdown and remained low during phase 2 to 4 (Figure 2, Table 1), despite increasing trends during intermediate phases and decreasing trends during lockdowns (Appendix A). Nevertheless, the percentage of contacts for RTIs with an antibiotic prescription slowly increased from the first lockdown on, but remained below the level of 2019. The number of patients with an antibiotic prescription for RTIs in the second intermediate phase, was still significantly lower than in that same period in 2019 (32.1 and 48.1/100,000 inhabitants per week in 2021 and 2019, respectively, Appendix A). 

In OOH-services, a similar temporary increase was seen in the contact rate for RTIs during the first lockdown (phase 1), followed by a decrease, after which the number of contacts remained lowered during phases 2 to 4 (Figure 3). Furthermore, prescribing rates were significantly lower during the first intermediate phase (phase 2) and remained lowered during the following phases (Table 1), despite an increasing trend during the first intermediate phase and decreasing trends during the first and second lockdown (Appendix A) The percentage of contacts with an antibiotic prescription was lower throughout 2020 and 2021, but temporarily increased during periods with lower contact rates. The antibiotic prescription rate was lower during the second intermediate phase, compared to the same period in 2019, before the COVID-19 pandemic (4.0 and 2.7 patients with a prescription for antibiotics for RTIs per 100,000 inhabitants in 2019 and 2021 respectively). 

### 2.3. Trends for Urinary Tract Infections

The average number of contacts for UTIs in DGPs (Figure 4) showed a different pattern than for RTIs, and was relatively constant over the years, apart from an initial decrease in the number of contacts during the first lockdown. This led to an increase in the percentage of contacts for UTIs with an antibiotic prescription, even though the average number of prescriptions per 100,000 inhabitants decreased significantly during the first lockdown compared to the pre-pandemic phase. Like the general antibiotic prescribing pattern in DGPs, antibiotic prescribing rates for UTIs increased significantly during intermediate phases (phase 2 and 4) and decreased during lockdowns (phase 1 and 3), compared to the previous phase (Table 1). During the second intermediate phase, the average number of patients with an antibiotic prescription per 100,000 inhabitants was at the same level as during that same period in 2019 (120.7 and 125.6 per 100,000 inhabitants in 2019 and 2021, respectively, Appendix A).

The average number of contacts for UTIs at OOH services (Figure 5) did not change during the different phases of the COVID-19 pandemic. However, antibiotic prescribing rates for UTIs did change temporarily in the second lockdown, with fewer prescriptions for antibiotics per 100,000 inhabitants (Table 1), despite an increasing trend in the first lockdown and a decreasing trend in the first intermediate phase (Appendix A). Prescription rates during the second intermediate phase for UTIs did not differ from prescription rates during the same period in 2019 (10.7 and 10.9 per 100,000 inhabitants in 2019 and 2021, respectively).

### 2.4. Trends for Different Age Categories

Figure 6 shows the percentage of contacts with an antibiotic prescription in DGP and OOH-services by age category. In DGP, antibiotic prescribing rates decreased during the first lockdown, in children aged 0–11 (from an antibiotic prescription in 5.8% of contacts in phase 0 to 3.9% of contacts during the first lockdown), in persons aged 12–74 (from 4.5% to 4.3%), and in older adults (75 and older, from 5.3% to 5.0%). The decrease was strongest in the youngest age group. In this age group, we also observed the strongest temporary increase in antibiotic prescribing during the second intermediate phase. 

At OOH-services the percentage of contacts with a prescription for antibiotics decreased slightly in persons aged 12–74 years (8.6% to 6.7%) and in older adults aged >74 years (8.1% to 6.8%). In young children aged 0–11 years, the decrease was more prominent: 7.0% to 4.6%. Furthermore, the decrease was preceded by an increase in the proportion of contacts with a prescription for antibiotics just before the start of the pandemic. During the second intermediate phase, the percentage of contacts with a prescription for antibiotics increased again. 

## 3. Discussion

GPs prescribed considerably fewer antibiotics during the COVID-19 pandemic in general and specifically for respiratory infections after lockdown restrictions became effective both in DGPs and OOH-services. After revoking restrictions, more antibiotics were prescribed, but the average prescribing rate in DGPs stayed lower than in 2019, while returning to pre-pandemic levels in OOH-services. In DGP, the antibiotic prescribing pattern seems to follow changes in the prevalence of RTI infections. However, in OOH-services, the decrease in antibiotic prescription rates overall, started at a later phase in the pandemic than the decrease in antibiotic prescriptions for RTIs. Moreover, the increase in prescriptions for RTIs at OOH-services seemed to commence later than the increase for all prescriptions. The steepest increases and decreases in antibiotic prescribing were observed in children both at DGP and OOH-services. 

Similar to studies in other countries, the Netherlands faced a decrease in general practice antibiotic prescribing during the first year of the COVID-19 pandemic [10,14,17]. This decrease can be partly explained by a decrease in contacts in general (both at the DGP and OOH-services) [28], for RTIs specifically [13,14,15,25,26] and most likely also for other infectious diseases such as gastrointestinal infections [15]. Patients were discouraged from contacting a doctor for mild RTI symptoms, and, from July 2020 on, were urged to get a COVID-PCR test through the Public Health Services first. The decrease in contacts for RTIs and other infectious diseases was also caused by a decrease of infections as a consequence of social distancing, closing of schools and day care, and hygiene measures like hand washing [6].

Nevertheless, the decrease in antibiotic prescribing for RTIs could not be fully explained by lower RTI contact rates alone. Patients who consulted the GP for an RTI (both DGP and OOH-services) were less likely to receive an antibiotic during the COVID-19 pandemic than before. It is likely that the case-mix of patients with respiratory symptoms consulting a GP changed, both as a result of government restrictions over time, but also with COVID-19 being the most prominent cause of a respiratory infection, for which antibiotics are not indicated [29,30]. A more in-depth study of antibiotic prescribing assessing specific RTI diagnoses and taking into account patient characteristics is required to assess what, besides lower contact rate for RTIs, caused the observed reduction in antibiotic prescribing for RTIs.

The decrease in antibiotic prescribing for UTIs was more modest than that for RTIs, as was the decrease in contacts for UTIs. This is not surprising, as the causal mechanism of UTIs is not related to COVID-19, nor to the restrictions and social distancing and hygiene advice. Notably, the steep decrease in DGP contacts for UTIs at the start of the first lockdown led to a temporal increase in antibiotic prescribing rate during UTI contacts at DGPs. It is likely that patients who contacted the GP for a UTI during that period were patients with a more severe UTI in which the likelihood of prescribing an antibiotic is higher.

We observed different patterns in antibiotic prescribing between DGP and OOH-services overall, but not for RTIs. The decrease in antibiotic prescriptions in general at OOH-services commenced during a later phase of the pandemic than in DGP. This finding may suggest undertreatment of infections in DGP, i.e., when patients wait longer before consulting their own GP. This may result in more severe infections that require urgent treatment out-of-hours, specifically for infections that were halted by lockdown measures. A previous study, however, did not find signals for more severe infections in DGP [15]. Contrary to the findings in our study, others reported a more immediate decrease in prescription rates of antibiotics at OOH-services, which may be related to a different organization of OOH-services in general and specifically for COVID-19 care [10,13].

As was observed in other studies, the decrease in prescribing was most prominent in children aged 0 to 11 years [17,26]. This is likely explained by lockdown measures, such as the closing of schools and day care facilities. Children generally have a high incidence of RTIs, for which antibiotics are regularly prescribed; the RTI incidence was greatly reduced during the COVID-19 pandemic. Furthermore, the prevalence of other diseases in children, such as acute otitis media and scarlet fever, decreased as well in the first months of the COVID-19 pandemic [31]. This also coincides with observations that the GP contact rates for children also showed the largest decrease [32]. The steepest increase in antibiotic prescribing when restrictions were revoked was also observed in children aged 0–11 years old.

### Strengths and Limitations of the Study

A strength of this study is that we used a nationally representative database, based on routine health data collected from electronic health records in DGP and OOH-services during the COVID-19 pandemic, and could compare our outcomes to pre-pandemic data from 2019. A limitation is that we did not have information on disease severity. Furthermore, the rate of RTI contacts may be an underestimation since during the first months of the pandemic patients with symptoms suggestive of COVID-19 were often seen at dedicated COVID-19 centers, sometimes situated at the locations of OOH-services. Not all visits to these centers may have been registered in the electronic health record system of the DGP or to a lesser extend at OOH-services. In addition, the electronic health records of DGPs may also include prescriptions from specialist or OOH-services. However, as the majority of antibiotics are prescribed by DGPs [33], we do not expect that this had a major impact on our results. A future study could link the data of OOH-services and DGPs at patient level to provide a comprehensive overview of antibiotic prescribing in general practice. Finally, we performed an interrupted time series analyses to assess changes in antibiotic prescribing between different phases of the COVID-19-pandemic. We were not able to correct for seasonal changes in antibiotic prescribing, as our period of data before the COVID-19-pandemic was relatively short compared to the period of the pandemic. Hardly any seasonal patterns were observed during the COVID-19 years 2020 and 2021. It is therefore likely that we underestimated the effect of the COVID-19 pandemic, particularly during winters, in which antibiotic prescribing is generally higher. 

## 4. Materials and Methods

### 4.1. Design, Setting and Database 

We performed a retrospective database study using routine care data from a minimum of 302 to a maximum of 379 DGPs and a minimum of 23 to a maximum of 28 OOH-services participating in Nivel Primary Care Database (Nivel-PCD). The number of DGPs and OOH-services varied between the weeks depending on quality and completeness of the data. DGP data covered approximately 8% of the Dutch population, OOH-services data about 60–70%. OOH-services in the Netherlands exclusively provide emergency primary care, not including hospital care. Both datasets included patient age, prescriptions (coded according to the Anatomical Therapeutic Chemical classification, ATC), contacts and diagnoses (International Classification of Primary Care-1 (ICPC-1) coded). Both DGPs and OOH-services were located throughout the country and their listed population was representative on age and sex for the Dutch population [32,34].

### 4.2. Data Analysis 

We analyzed the number of patients with at least one systemic antibiotic prescription (ATC-code J01) per week per 100,000 listed inhabitants, for all inhabitants for 2020 and 2021, compared to 2019. In addition, the percentage of contacts with an antibiotic prescription per age category (0–11, 12–74, 75 and older) was calculated. Per age category we divided the number of contacts with an antibiotic prescription by the total number of contacts per week in that age category. We also studied the two most common diagnoses for which antibiotics are prescribed: respiratory tract infections (RTI, ICPC codes R74—acute upper respiratory infection, R75—acute/chronic rhinosinusitis, R77—acute laryngitis/tracheitis, R78—acute bronchitis/bronchiolitis, R80—influenza, and R81—pneumonia) and urinary tract infections (UTI, ICPC-code U71). As contact rates changed during the COVID-19 pandemic, also the number of contacts with and without an antibiotic prescription for RTIs and UTIs, respectively, was calculated per 100,000 listed inhabitants. In addition, we calculated the percentage of RTI and UTI contacts in which an antibiotic was prescribed. A three-week moving average was calculated to correct for any fluctuations due to, e.g., holidays. 

Next, we studied the changes in antibiotic prescribing between the five different phases of the pandemic (see Box 1) using one-way ANOVA analyses with a post hoc Bonferroni multiple-comparison and interrupted time series analyses (ITSA). ANOVAs were performed to study whether there were differences in the average antibiotic prescribing rates between phases. ITSA was used to study whether there were differences in antibiotics rate (i.e., slope of the regression line) between the different phases. Standard errors were corrected for the autocorrelation in the time series. For both analyses, each phase was compared to the previous phase. In addition, to determine whether the level of antibiotic prescribing during phase 4 returned to the level of antibiotic prescribing in 2019, we performed a two-sided t-test in which we compared the mean incidence of antibiotic prescribing between phase 4 of the pandemic in 2021 with the same period in 2019. Analyses were performed using Stata 16. 

### 4.3. Ethics

The study was approved by the relevant governance bodies of Nivel-PCD under number NZR-00320.056. According to Dutch legislation, and under certain conditions, neither obtaining informed consent nor approval by a medical ethics committee is obligatory for this kind of observational studies [35,36,37].

## 5. Conclusions

The restrictions taken to combat the COVID-19 pandemic caused a decrease in antibiotic prescribing in the Netherlands, both in DGP and OOH-services. The decrease does not seem permanent, as antibiotic prescriptions increased again when government restrictions were revoked and the prevalence rate of RTIs increased. The temporary decrease in prescriptions is partially explained by a decreased number of infections and consultations. However, it remains unclear how prescriptions rates were impacted by undertreatment, fewer unnecessary prescriptions, remote consultations and prevention of infections by, e.g., increased hand washing. More insight in these aspects is necessary to learn from the COVID-19 pandemic for future endeavors aimed at decreasing prescription rates for antibiotics. 

## Figures and Tables

**Figure 1 antibiotics-11-00309-f001:**
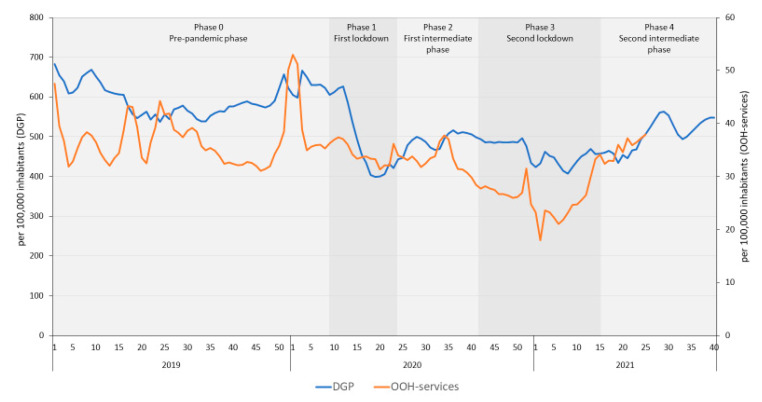
Number of patients with an antibiotic prescription during daytime- and out-of-hours general practice, per 100,000 inhabitants per week from 2019 to 2021 (week 40 for DGP and week 25 for OOH-services). DGP = daytime general practices; OOH-services = out-of-hours services.

**Figure 2 antibiotics-11-00309-f002:**
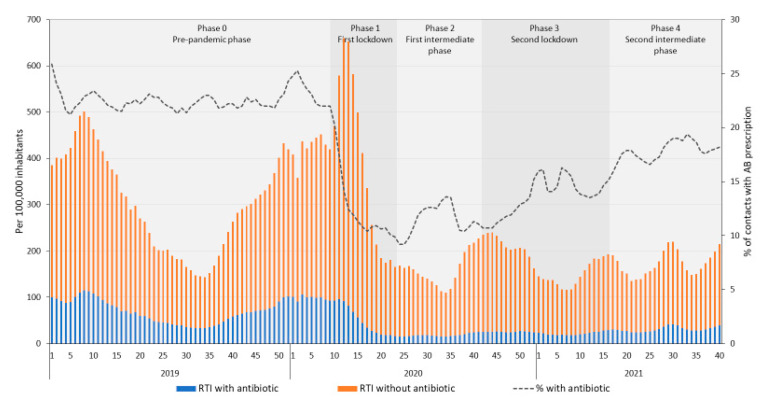
Number of contacts for respiratory tract infections (RTIs) with and without an antibiotic prescription in daytime general practice, per 100,000 inhabitants per week and the percentage of RTI contacts with an antibiotic prescription from 2019 to 2021 week 40.

**Figure 3 antibiotics-11-00309-f003:**
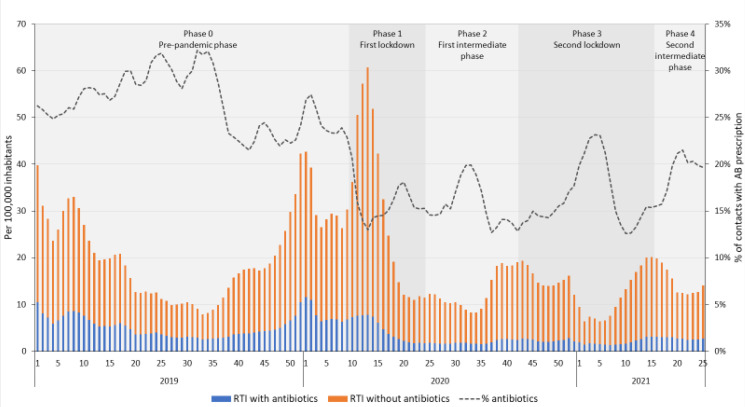
Number of contacts for respiratory tract infections (RTIs) with and without an antibiotic prescription in out-of-hours services, per 100,000 inhabitants per week and the percentage of RTI contacts with an antibiotic prescription from 2019 to 2021 week 25.

**Figure 4 antibiotics-11-00309-f004:**
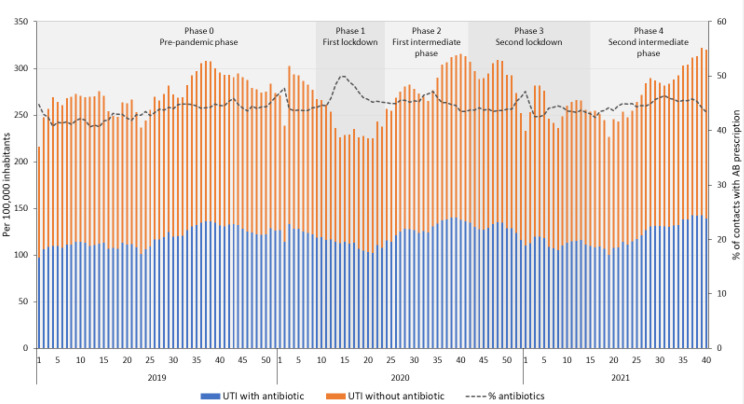
Number of contacts for urinary tract infections (UTIs) with and without an antibiotic prescription in daytime general practice, per 100,000 inhabitants per week and the percentage of UTI contacts with an antibiotic prescription from 2019 to 2021 week 40.

**Figure 5 antibiotics-11-00309-f005:**
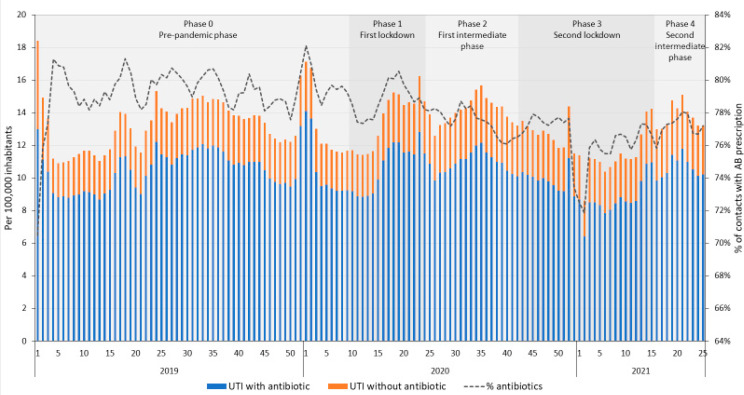
Number of contacts for urinary tract infections (UTIs) with and without an antibiotic prescription in out-of-hours services, per 100,000 inhabitants per week and the percentage of UTI contacts with an antibiotic prescription from 2019 to 2021 week 25.

**Figure 6 antibiotics-11-00309-f006:**
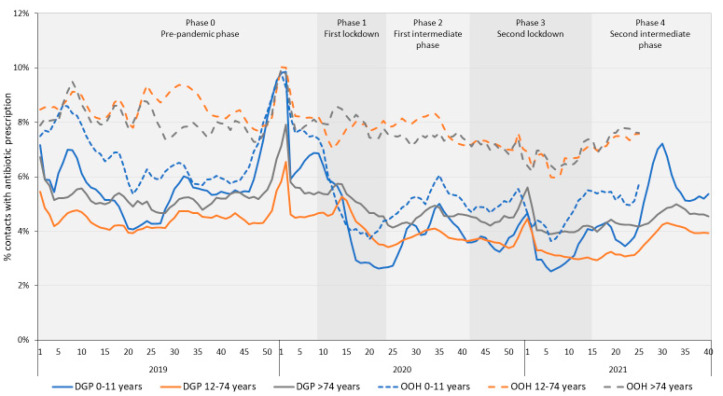
Percentage of daytime and out-of-hours general practice contacts with an antibiotic prescription by age category, per week from 2019 to 2021 (week 40 for DGP and week 25 for OOH-services). DGP = daytime general practice, OOH = out-of-hours services.

**Table 1 antibiotics-11-00309-t001:** Number of patients with an antibiotic prescription per week per 100,000 inhabitants for the different phases of the COVID-19 pandemic.

	DGP			OOH-Services		
	Mean	SD	*p*-Value for Difference with the Previous Phase *	Mean	SD	*p*-Value for Difference with the Previous Phase *
**Overall incidence of antibiotic prescriptions**						
Phase 0 (pre-pandemic)	595.160	39.066		36.162	7.539	
Phase 1 (first lockdown)	492.190	89.872	<0.001	34.210	3.356	1.000
Phase 2 (first intermediate phase)	491.812	18.996	1.000	32.403	3.693	1.000
Phase 3 (second lockdown)	459.830	26.314	0.195	26.240	5.947	0.008
Phase 4 (second intermediate phase)	510.815	39.772	<0.001	36.610	3.183	0.012
**Incidence of antibiotic prescriptions for RTIs**						
Phase 0 (pre-pandemic)	72.810	25.315		5.233	2.499	
Phase 1 (first lockdown)	49.981	32.308	0.001	4.196	2.669	0.712
Phase 2 (first intermediate phase)	18.135	3.214	<0.001	1.985	0.475	0.005
Phase 3 (second lockdown)	23.354	3.222	1.000	2.127	0.692	1.000
Phase 4 (second intermediate phase)	32.100	6.657	1.000	2.747	0.370	1.000
**Incidence of antibiotic prescriptions for UTIs**						
Phase 0 (pre-pandemic)	119.272	10.052		10.376	2.144	
Phase 1 (first lockdown)	111.606	5.414	0.069	10.677	1.778	1.000
Phase 2 (first intermediate phase)	129.900	7.416	<0.001	10.943	0.745	1.000
Phase 3 (second lockdown)	119.500	9.627	0.009	9.172	2.148	0.042
Phase 4 (second intermediate phase)	125.596	13.008	0.258	10.871	1.134	1.000

Abbreviations: DGP = daytime general practice; OOH = out-of-hours; SD = standard deviation. * Outcomes of one-way ANOVA analyses with post hoc Bonferroni multiple-comparison test, differences between phases are presented if the overall effect of the phases was significant. *p*-value for comparison of phase with previous phase.

## Data Availability

Not applicable.

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
