# Peer review of "Antibiotic Prescribing in Dutch Daytime and Out-of-Hours General Practice during the COVID-19 Pandemic: A Retrospective Database Study"

_antibiotics, 2022, doi:10.3390/antibiotics11030309_

Round 1

Reviewer 1 Report

The reviewer has the rare pleasure to read a very dignified, clear, concise, and precisely written manuscript and to state with pleasure that in that manuscript he had almost nothing to suggest / change. A few small changes that are insignificant and that even if they remain as they are, would not jeopardize the strength of the manuscript.

Line 246. Please, put “otitis media” in italic style

Subtitle 3.1. Please write: strengths and limitations of this study

Lines 283-285 This statement is not clear (We performed a retrospective database study using routine care data from 302 to 379 283 DGPs and 23 to 28 OOH-services participating in Nivel Primary Care Database (Nivel-284 PCD).

What is number 302 and what represents the number 379? is that the DGP services number? if in the examined time frame, data from all DGPs were not used in all time periods, the authors are asked to write precisely. for example, in the first 3 months of 2020 we used data from 313 DGPs, and in the next 3 months from 344 DGPs, etc. And if that's not the case, the authors are asked to explain what it is. The same with the number of OOH

It is unclear whether hospital cases of antibiotic prescribing were included in the study. Do they belong to OOH? the reviewer thinks they do not belong, but it is not entirely clear. The authors are asked to write in the introduction or material and methods (NOT in the limitations of the study, because it is not limitation) in one or two sentences whether OOH includes some specialized hospitals (surgery, orthopaedics, oncology). therefore, it is necessary to see whether specialized hospitals with inpatients were in any way involved or not in this study.

Please, write in the legend of all figures was a timeline that includes data from figures (from-to).

Reviewer 2 Report

ï‚· A brief summary
The aims of this study are to describe changes in antibiotic prescribing in the different phases of the COVID-19 pandemic in 2020 and 2021 compared to 2019 in DGPs and at OOH-services and to investigate RTIs and UTIs treatment separately. The strength of the study is the data source, with weekly data and detailed patient information.

I recommend the manuscript for publication, after a minor revision.

ï‚· General concept comments
Article:
The manuscript deals with a current issue, using a national database. The conclusions are moderate and appropriate. The authors evaluate the results considering the limitations.

ï‚· Specific comments.
Why were the DGP and OOH antibiotic prescriptions not examined together by the working group? If we were to sum these two sectors (DGPs + OOH), would the total
give a similar result?
line 258-260:
Please, estimate the percentage that is not recorded electronically.
Method section:
line 293-295: Please clarify the % of contacts calculation.
General questions to help guide your review report for research articles
ï‚· Is the manuscript clear, relevant for the field and presented in a well-structured manner?
Yes
ï‚· Are the cited references current (mostly within the last 5 years)? Does it include an abnormal number of self-citations?
The cited references are relevant and mostly from the last 2 years, and there are only a few self-citations.
ï‚· Is the manuscript scientifically sound and is the experimental design appropriate to test the hypothesis?

Yes
ï‚· Are the manuscript’s results reproducible based on the details given in the methods section?
Yes
ï‚· Are the figures/tables/images/schemes appropriate? Do they properly show the data?
Are they easy to interpret and understand? Are the data interpreted appropriately and consistently throughout the manuscript? Please include details regarding the statistical analysis or data acquired from specific databases.
Tables:
The tables in the manuscript are appropriate.
Figures:
The figures in the manuscript are correct and appropriate, however, the labels, legends
and axes are barely visible and need to be readable.
ï‚· Are the conclusions consistent with the evidence and arguments presented?
Yes
ï‚· Please evaluate the ethics statements and data availability statements to ensure they are adequate.
They are adequate.
